# Effect of Electrode Profile and Polarity on Performance of Pressurized Sparkgap Switch

Vinod Kumar Gandi [1,2,*], Rishi Verma [1,2], Manoj Warrier [1,3] and Archana Sharma [1,2]

1.  Homi Bhabha National Institute, Training School Complex, Anushakti Nagar, Mumbai 400094, India; rishiv@barc.gov.in (R.V.); manoj@barc.gov.in (M.W.); arsharma@barc.gov.in (A.S.)
2.  Pulsed Power & Electromagnetics Division, Bhabha Atomic Research Centre Facility, Visakhapatnam 531011, India
3.  Computational Analysis Division, Bhabha Atomic Research Centre Facility, Visakhapatnam 531011, India
*   Correspondence: gvinod@barc.gov.in

**Abstract:** Sparkgap are most widely used closing switches in various high-voltage pulsed power systems and its reliable operation at desired voltage level is very essential. Conventionally by adjusting the filling gas pressure inside sparkgap switch, breakdown voltage level is altered but switching characteristics such as stability in hold-off voltage at various pressures, breakdown delay, plasma channel formation, and erosion rate are mainly dictated by adopted electrode profile and its dimensions, inter-electrode gap length and polarity. In this paper, experimental results obtained on breakdown characteristics of four different electrode geometries—Plane Parallel, Hemi-spherical, Bruce, and Rogowski and also a generalized criterion for fixing major dimensions of electrode and inter-gap length to ensure uniform electric field in the inter-electrode region are reported. All electrodes are of brass material and have common radius and thickness of 25 mm and 18 mm, respectively (surface finish <1 μm). Experiments performed on various electrode profiles in gap lengths of 2 mm to 5 mm range with pure nitrogen ($N_2$) gas pressurization up to 50 psi reveal that among all profiles, Rogowski performs most reliably having stable hold-off voltage in wide operating range. Hold-off voltage magnitude and breakdown delay was commonly obtained higher for negative polarity in all trials. A comprehensive overview of experimental investigation reported herein compares suitability of various electrode profiles and polarity for reliable switching.

**Keywords:** sparkgap; electrode profile; Rogowski; Bruce; plane parallel; hemi-spherical; uniform electric field; polarity; breakdown delay



## 1. Introduction

In pulsed power technology the generic capacitor energy storage scheme consists of voltage source, capacitor, load, and intermediate switch [1]. Ignitron, Pesudospark, Sparkgap, Trigatron, and Railgap are the commonly used closing switches in pulsed power systems [2]. Among these, sparkgap switch is most widely used due to its simple construction, ease of operation, and being an economical alternative. The most desired feature in sparkgap switch is that it must have stable hold-off in wide operating voltage range and offers minimum inductance. Though in most of the pulsed power systems working voltage ranges between 5 kV and 50 kV, sparkgap switches are needed to be designed in such a way that they are able to hold high voltage in a very small discharge volume to ensure minimum inductance. In such a scenario, electrode profile plays a key role in counter balancing the hold-off voltage and switch inductance.

Electrode profile dictates the spatial distribution of electric field (E) in the gap resulting in the formation of uniform and non-uniform field in the inter-gap region. The E due to infinite parallel plate capacitor is a classic example of an ideal uniform field but its realization is difficult in real high voltage systems. In practical systems, uniform fields are realized by profiling and arranging electrodes in specific geometries like Plane-parallel,

Bruce, Rogowski, Chang, Ernst [3–5] etc. In practical uniform field gap, electric field is more in the discharge region and minimum as it goes toward the edges. In recent times these specially contoured electrode profiles are widely used in high power and high repetitive transversely excited gas lasers [6–8].

The breakdown mechanism in the uniform field gaps is explained by streamer theory. According to the streamer theory, transition phase from primary avalanche to cathode or anode-directed streamer can take place if it satisfies any of these following conditions: (i) As space charge electric field reaches applied "*E*"; (ii) as primary avalanche size reaches "critical size" $N_{ecr}$ i.e., ~$10^7$ electrons; and (iii) the transition may also occur as the avalanche passes the critical distance = ln ($N_{ecr}$) /$\alpha$ (where "$\alpha$" is ionization coefficient) [9–12]. Hadeer Hassan et al. [13] simulated the avalanche evolution and its transition into anode and cathode-directed streamer in short uniform field gaps. The influence of electrode configuration on the breakdown voltage subjected to different non-uniform field conditions is studied by Zhang et al. and Sham et al. [14,15].

In non-uniform fields there is absolute spatial variation of *E* in discharge gap and dominant space charge effect that results in localized breakdown due to dense electric stresses [16]. The breakdown mechanism in the non-uniform field gaps is explained by corona stabilization effect. This effect is due to the accumulation of space charge in the discharge gap. The space charge behavior is mainly influenced by the polarity of applied voltage, and so the breakdown mechanisms behind the switch operation are different for different polarities of applied voltage. The difference between positive and negative polarity self-breakdown-voltage (SBV) depends upon the degree of field non-uniformity, nature of gas, and operating pressure [17–20]. L. Li et al. and Hogg et al. [21,22] studied the effect of polarity on breakdown voltage magnitude by considering the distribution of space charge for positive and negative polarity applied voltage.

The breakdown characteristics for non-uniform field conditions does not follow the typical Paschen curve, instead the breakdown curve has a point of singularity at critical pressure($p_c$), below the critical point the curve follows the increasing trend and trailed by decreasing trend at slower rate as pressure exceeds $p_c$. Mardikyan et al. [23] reported that at high pressures the 50% positive impulse breakdown voltage in 1%: 99% $SF_6$/dry air is greater than 100% pure $SF_6$ as corona stabilization effect in dry air is extended to higher pressure as compared to pure $SF_6$ due to immobile $SF_6$ ions. The switches operating by means of corona stabilization effect are termed as corona-stabilized switches and they exhibit higher recovery rate as compared to conventional sparkgap switches [24–27].

Apart from source and load parameters, the rise time of the output pulse mainly depends on plasma channel inductance *(L(t))* of the sparkgap switch. The parameter *L(t)* is functionally dependent upon gap length (d) and discharge channel radius ($k_z(t)$)) [28]. The approximate relation for modelling the discharge channel width = $\sqrt{2D(t)t}$ (where *<D(t)>* is the electron diffusion coefficient [29]. From the theoretical calculations and numerical simulations of streamer model within 2D approximations the discharge channel width was considered to be constant [30]. As a result, electrode profile may have a negligible effect on the rise time of output pulse since $k_z(t)$) was constant. One way to reduce the plasma channel inductance is by having a multiple channel discharge instead of single channel. The other advantage of multi-channeling is that it results in the uniform electrode erosion thereby increasing the life span of the switch. Bindu et al. [31] compared the output pulse rise time with two different electrode geometries via electrodynamic simulations. The breakdown delay times ($t_{bd}$) and time jitter ($\sigma_{bd}$) are the two prior characteristics parameters that play a pivotal role when synchronization of multiple events is involved. Rishi et al. [32] reported that breakdown delay times can be reduced by (a) operating near to SBV and (b) using trigger pulse with higher *dV/dt*.

The effect of electrode profile on switching reliability and hold-off voltage magnitude in uniform field conditions has not been collectively addressed yet in the literatures and this gap has been thoroughly explored in this reported study. The effect of polarity on

breakdown delay times due to space charge is a novel observation seen in this experimental study.

In this work, electrostatic simulations and breakdown characteristics for various uniform electrode profiles are presented. The simulation results provide a reference for fixing the electrode dimensions in order to avoid the edge effect, which results in premature breakdown. For the subjected operational constraints such as voltage, gap length, the experimental results provide a reference to choose an electrode profile, which results in a stable breakdown behavior and uniform electrode erosion.

## 2. Materials and Methods

### 2.1. Test Electrode Configuration

In this reported work, characteristics study of four most commonly used electrode configurations in sparkgap switches i.e., Plane-parallel, Rogowski, Bruce, and Hemi-spherical electrode profile is presented. The fabricated pairs of these electrode geometries are shown in Figure 1.

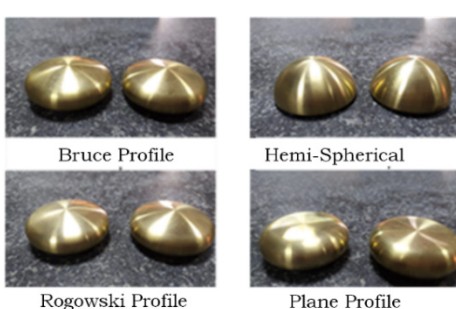

**Figure 1.** Photograph of various electrode pair configurations.

The design and constructional schematic of these four electrode geometries are illustrated in Figure 2. The complete electrode geometry is shaped by rotating the given profile around its axis of symmetry. $A$ is characteristic distance, $\alpha_o$ is characteristic angle, and $r_e$ is the radius of circular section. Bruce and Rogowski have complicated profiles and require programmed machining (stepwise procedure and equations involved in designing of these profiles are detailed in [33,34]).

The physical dimension of fabricated electrode profiles, Plane, Rogowski, Bruce, and Hemi-sphere is shown in Figure 3.

Rogowski proposed an electrode configuration based on analytic function given as [3]–

$$Z = (A/\pi) \times (\omega + e^{\omega}) \tag{1}$$

Here, $Z = X + iY$ and $\omega = \phi + i\psi$. The parameters $\phi$, $\psi$ indicate the line of force and equipotential surface, respectively. The conventional Rogowski profile corresponds to the equipotential surface $\psi = 90°$ and the conformal mapping of Equation (1) for $A = \pi$ is shown in Figure 4.

The electric field and equipotential lines between the electrodes shown in the conformal mapping in Figure 4 clearly indicate that a nearby uniform field exists in Region-1, where the discharge occurs and it is the region of maximum electric field $E$. As we transit from the Region-1 to Region-2, the electric field becomes weaker and non-uniform.

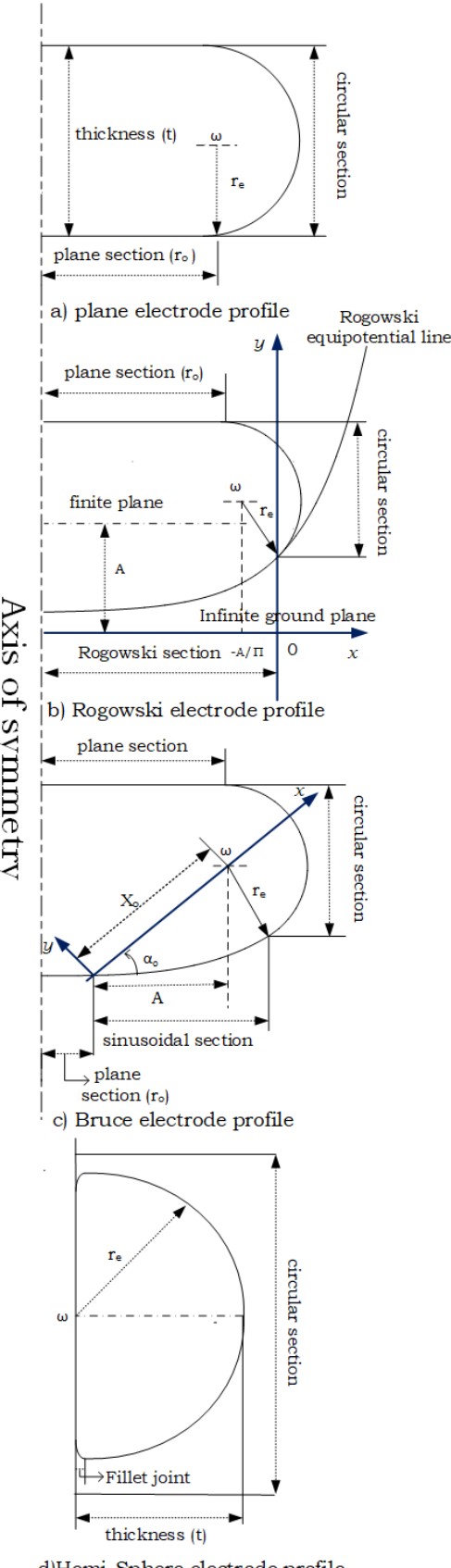

**Figure 2.** Constructional design schematic of (**a**) Plane, (**b**) Rogowski, (**c**) Bruce, and (**d**) Hemi-sphere electrode profile.

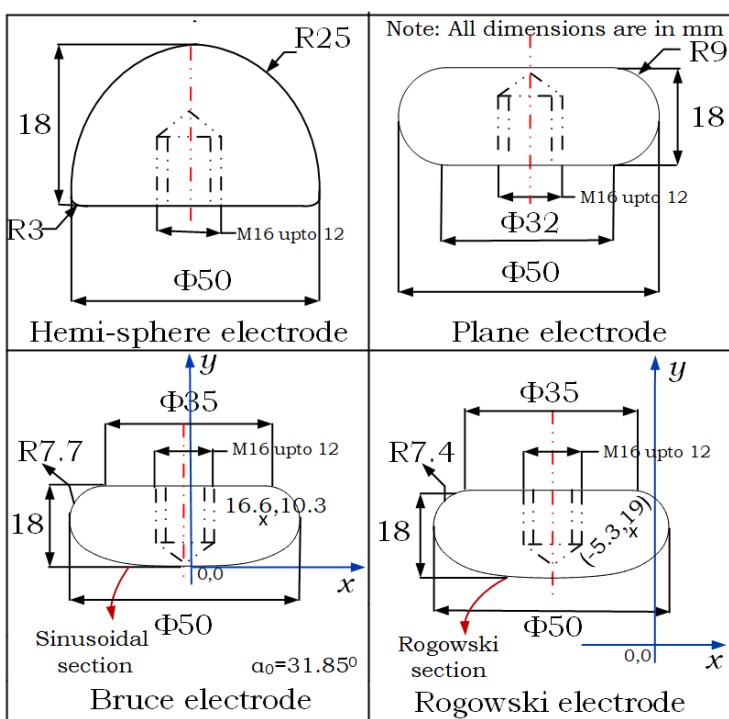

**Figure 3.** Physical dimensions of fabricated electrodes.

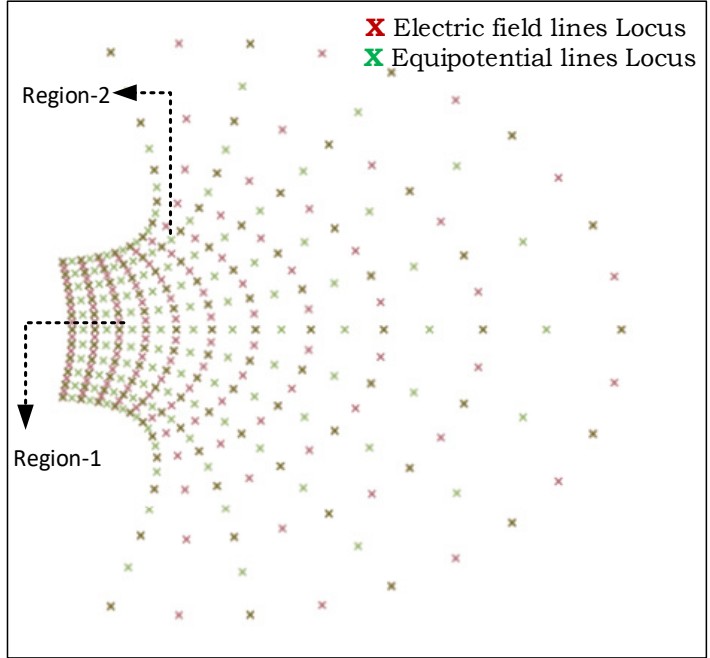

**Figure 4.** Conformal mapping of Rogowski profile for A = π.

### 2.2. Simulation Results and Analysis

Simulations have been performed to analyze the static fields of different electrode geometries. The analysis of static electric field *E* is the pivotal parameter in sparkgap switch design as the switch behavior in steady state depends on the static E gradient contour. The static E field calculations have been performed using the relation

$$\nabla^2 V = \left( \frac{\partial^2}{\partial x^2} + \frac{\partial^2}{\partial y^2} + \frac{\partial^2}{\partial z^2} \right) V = 0 \tag{2}$$

The tangential and normal boundary conditions at material interface are given as

$$n.(D_2 - D_1) = \rho_S \qquad n \times (E_2 - E_1) = 0 \qquad (3)$$

Finite element method (FEM), charge simulation method, boundary element method, and finite difference method are different solution methodologies [16] to solve Equation (2). In this study, electrostatic field simulations are carried out using CST Studio® [35]. It is remarkable to note that steady state behavior of sparkgap switch critically depends on two major physical parameters i.e., inter-electrode gap length *d* and electrode thickness *t* as shown in Figure 5.

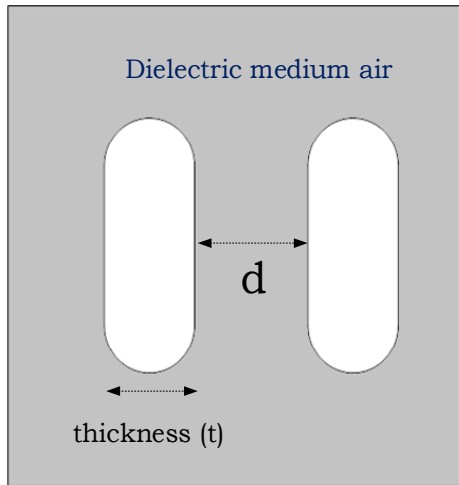

**Figure 5.** Basic model of two electrode sparkgap switch.

The electrical breakdown strength (kV/cm) of air in uniform field gaps is given as:

$$E = 24.6p + 6.7\sqrt{\frac{p}{d}} \qquad (4)$$

where, *p* is the filling gas pressure (in bars) inside sparkgap switch and *d* is the inter-electrode gap length (in cm). The above relation Equation (4) is valid over the *pd* range of $10^{-3} \leq pd \leq 50$ bar-cm. The modified relation considering the effect of geometrical field enhancement [36] is given as:

$$E = 24.6p + 6.7\delta\sqrt{\frac{p}{d_{eff}}} \qquad (5)$$

The factor $\delta$ (delta) in Equation (5) is a function of *d/t* in the case of uniform field electrodes. The spatial *E*-field distribution obtained for different *d/t* ratios is shown in Figure 6.

In this work, thickness *t* of all electrode profiles is kept fixed at 18 mm and relative changes in the *E* field distribution are analyzed by varying the inter-electrode gap length *d* as 9 mm, 10.8 mm, 14.4 mm, and 18 mm (schematic shown in Figure 5). Analytical results shown in Figure 6 infer that for the case $d/t \leq 0.5$, the locus of *E(x)* is straight line indicating uniform field condition whereas for the case $d/t > 0.5$, *E* is higher on electrode surface as compared to center. The edge effect and *E* focusing on electrode surface may result in pre-firing of the switch. The E-field distribution for Plane, Bruce, Rogowski, and Hemi-sphere electrode profiles simulated in CST Studio ® for the cases $d/t < 0.5$ and $d/t > 0.5$ (for applied voltage of 10 kV DC) is shown in Figure 7.

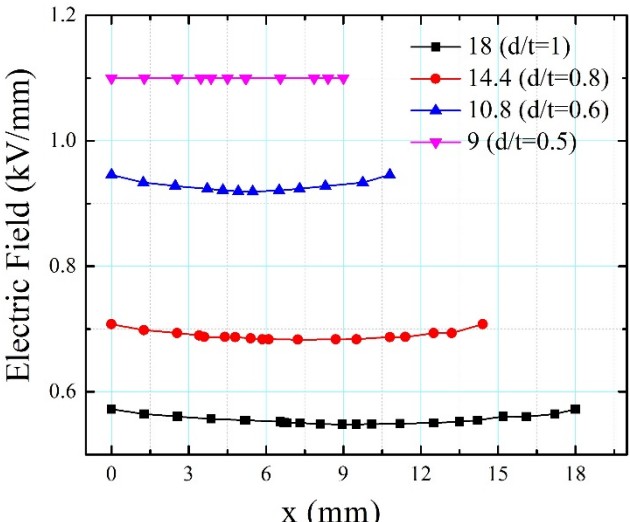

**Figure 6.** Spatial variation of *E(x)* in gap for different *d/t* ratios (where *x* is any point in the inter-electrode gap).

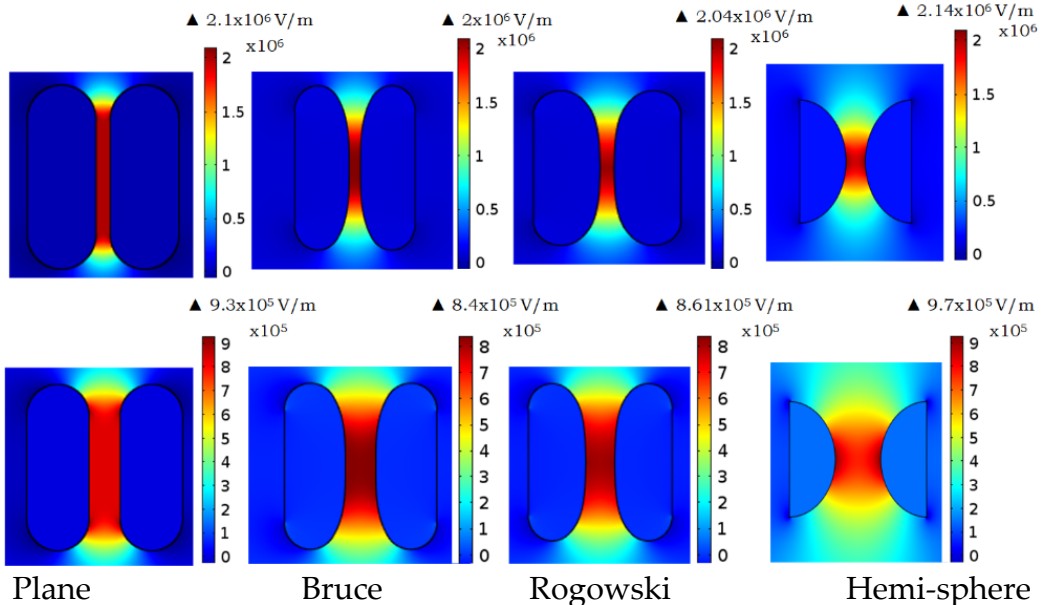

**Figure 7.** *E* distribution for electrode profiles for the case *d/t* < 0.5 (**upper**) and *d/t* > 0.5 (**bottom**).

Electric field distribution shown in Figure 7 for various electrode profiles clearly indicates that for the case *d/t* < 0.5 (i.e., upper row) uniform field region and *E*-field is more concentrated in the central region whereas for the case *d/t* > 0.5 (bottom row) *E*-field diverges radially outward and its concentration increases at the edges (which is detrimental for stable steady state breakdown characteristic of switch). The relatively minimal change in E-field distribution for the case *d/t* > 0.5 or < 0.5 in Hemi-sphere electrode profile infers stable breakdown behavior in wider range of inter-electrode gap lengths.

The performance index to evaluate the field uniformity is given as:

$$\eta = \frac{E_{max}}{E_{avg}} \tag{6}$$

Here, $E_{avg} = V/d$ and $E_{max}$ is maximum electric field in gap. The field uniformity factor *η* versus gap length *d* for different electrode profiles is shown in Figure 8.

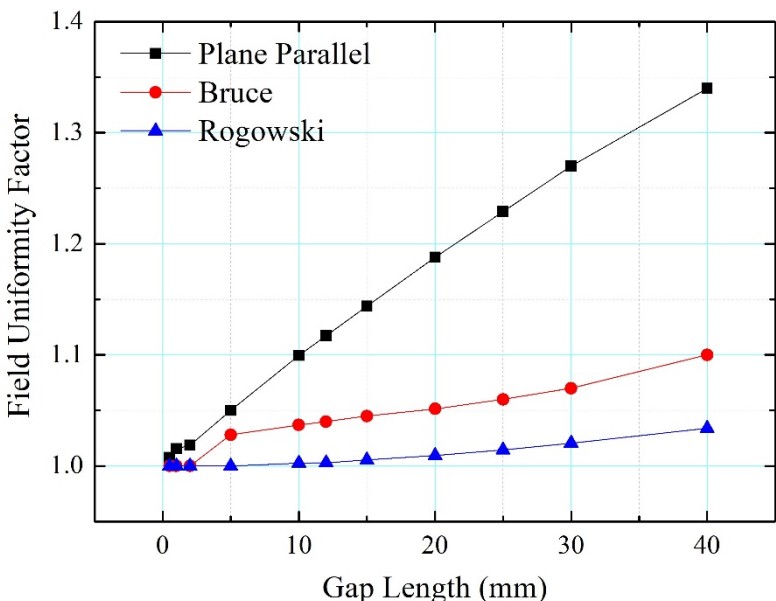

**Figure 8.** η versus gap length *d* for different electrode profiles.

Graph shown in Figure 8 illustrates the dependence of field uniformity factor $\eta$ on electrode profile and inter-electrode gap length *d*. For the stable steady state breakdown characteristics, $\eta$ should remain around unity for extended gap lengths for e.g., in Rogowski profile electrode there is a very marginal increase in $\eta$ value with subsequent increase in inter-electrode gap length (conversely plane parallel electrode profile configuration may be considered most unstable at higher gap lengths; $\eta$ slope for Bruce profile moderately increases with gap length). The field-enhancement-factor (*FEF*) for hemi-sphere electrode profile is given as [37]:

$$FEF = \frac{0.9}{r} \times \left( r + \frac{d}{2} \right) \tag{7}$$

Here *d* is the inter-electrode gap length and *r* is the radius of hemi-sphere profile. In gas breakdown Equation (5) $d_{eff}$ is a function of geometry and it is 0.115r for hemi-spherical profile. Estimation of factor $\delta$ in Equation (5) was done by curve fitting plot of Giddings model that is given as:

$$\delta = y_o + \frac{A}{W} \times \left( \sqrt{\frac{x_c}{x}} \right) \times I_1 \left( \frac{2\sqrt{x_c \times x}}{w} \right) \times \exp\left( \frac{-(x + x_c)}{w} \right) \tag{8}$$

Here the values of $y_0$, $x_c$, *W*, and *A* are 0.98, 0.56, 0.078, and $-0.18$ respectively. $I_1$ is the modified Bessel function of first kind. The operating region where this experimental work has been carried out with hemi-spherical geometry is highlighted in Figure 9.

Graph shown in Figure 9 infers that to get a high degree of field uniformity in hemi-sphere electrode profiles it is advisable to maintain the *d/r* ratio > 1.

### 2.3. Experimental Setup and Diagnostics

A demountable and refurbishable type coaxial sparkgap switch assembly has been made and directly mounted over the 1 μF/100 kV energy storage capacitor as shown in Figure 10. For minimizing connection inductance, integration of sparkgap switch with capacitor is done in squirrel cage configuration.

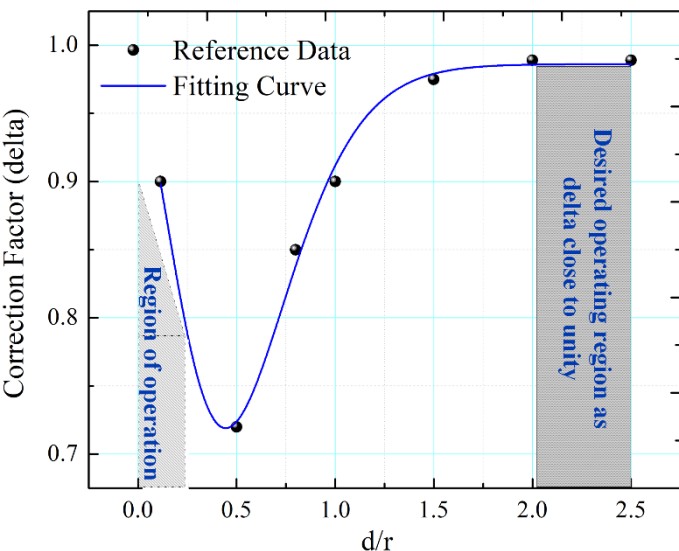

**Figure 9.** Correction factor δ Vs *d/r* for hemi-sphere profile.

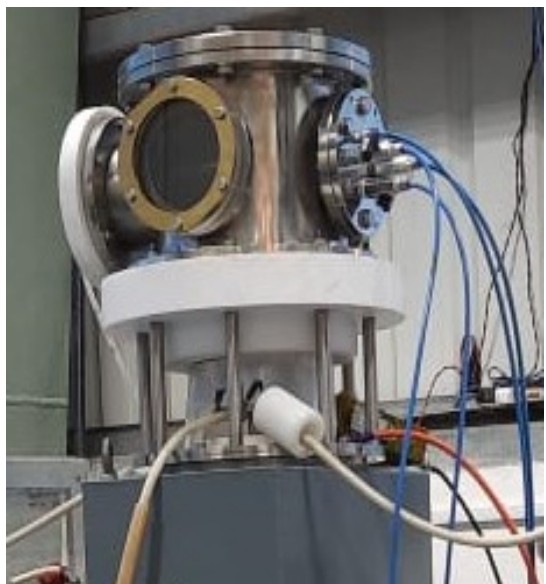

**Figure 10.** Sparkgap characterization test bench setup.

The sparkgap chamber is made of SS-304 material and it has diameter and height of 205 mm and 115 mm, respectively. This chamber can be safely pressurized up to 150 psi and it is capable of withstanding high pressure shock waves generated during high charge transfer discharge [38]. A 100-mm thick Teflon flange was used for connecting high voltage electrode. The discharge electrode assembly arrangement made inside this switch facilitates integration and replacement of any electrode profile as per requirement. With the help of check-nuts, the inter-electrode gap length may be adjusted from 1 mm to 40 mm. Arrangement for fixing an intermediate trigger electrode has also been facilitated. Constructional schematic of this coaxial sparkgap switch assembly is shown in Figure 11.

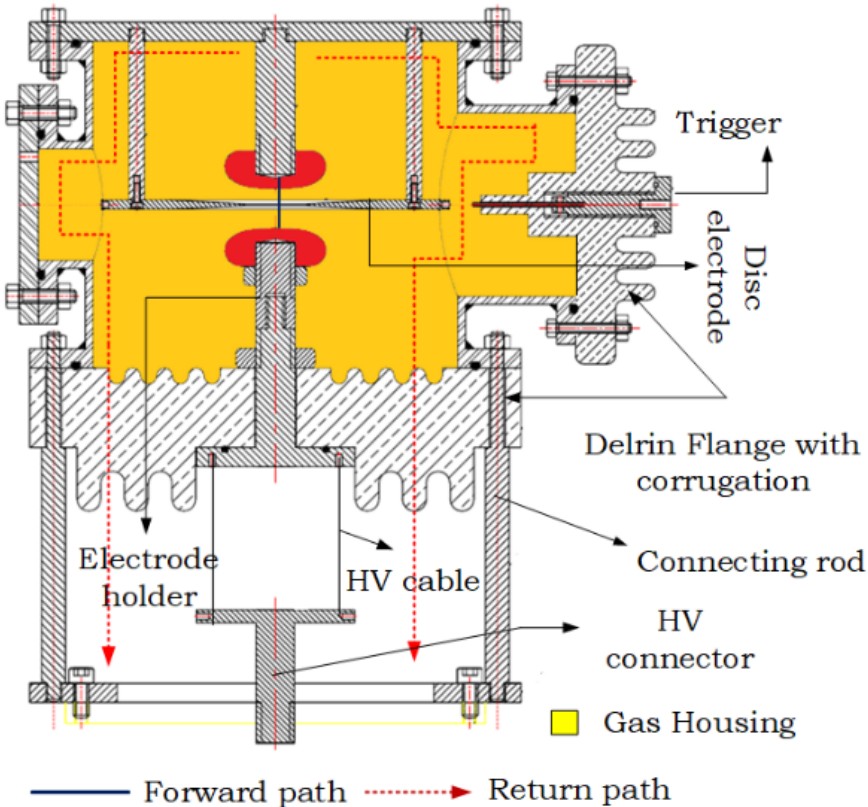

**Figure 11.** Constructional schematic of coaxial sparkgap switch.

For measuring high voltage up to 40 kV DC, HV probe #Zeonics-40/28 was used. A standard 230 kV, 20 MHz bandwidth probe (#NorthStar-VD150) was used for the measurement of fast high voltage pulses. For measuring fast current pulse up to 200 kA, a standard clamp-on CT of sensitivity 0.001 V/A (#Pearson-5664) was used. Piezo-resistive transducer #MICRON-P8700 was used for the precise measurement of filling gas pressure inside the switch. The data acquisition system consists of digital storage oscilloscope Agilent DSO7104B (4 GS/s, 1 GHz). Double shielded coaxial cables were used for signal transport to avoid external electromagnetic noise interference during the experimental run.

## 3. Results and Discussion

For the purpose of characterizing breakdown behavior of various electrode profiles, experiments were conducted on the test bench setup as shown in Figure 10. Inter-electrode gap length was varied in the range of 2 mm to 5 mm and discharges were performed with nitrogen ($N_2$) gas pressurization in the range of 5 psi to 50 psi. In a set, data for ten successive shots were collected at each pressure setting and inter-electrode gap length for the respective electrode profiles. The self-breakdown-voltage (SBV) measurements were done for both positive and negative bias with respect to ground electrode for four different electrode profile.

The sample waveform of current and voltage discharge signal captured in the event of self- breakdown in sparkgap switch is shown in Figure 12. It may be noted from the under-damped current discharge trace that time period of discharge signal is ~3.1 μs and this infers a residual circuit inductance of ~244 nH.

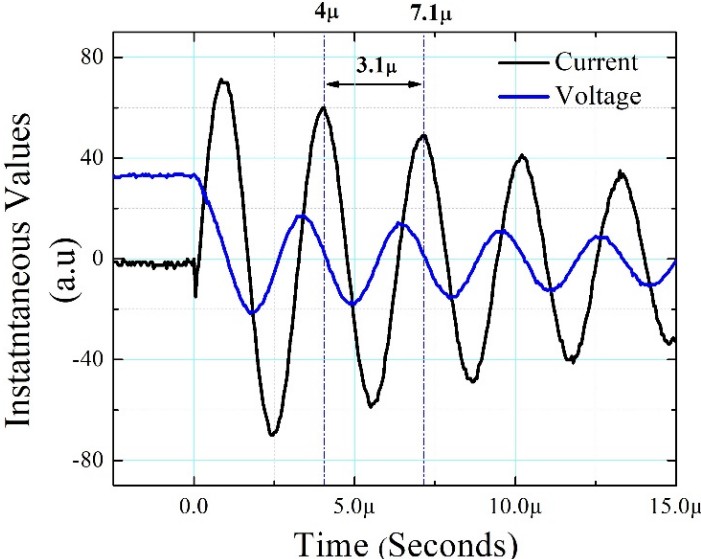

**Figure 12.** Current and Voltage discharge waveform.

The experimental results obtained for Plane Parallel and Hemi-sphere electrode profiles are shown in Figure 13a,b.

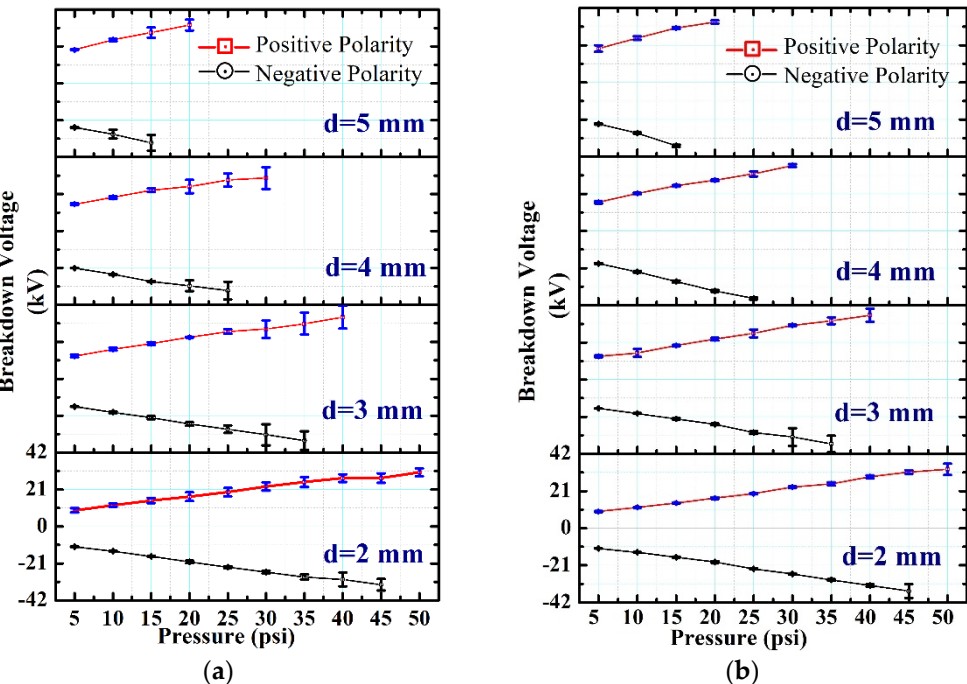

**Figure 13.** Results of Plane-Parallel and Hemi-sphere profiles. (**a**) Plane Parallel. (**b**) Hemi-Sphere.

It may be noticed from below graphs that mean value of SBV and standard deviation (SD) varies as a function of *pd*. It may also be noted that in subsequent shots at high voltage/pressure, significant variation in SBV was observed and it has been denoted as factor Δ (it basically denotes variation in SBV of switch with specific electrode profile at higher voltage/pressure).

In the case of plane-parallel electrode profile at voltage below 30 kV, Δ is typically ~6 to 7 kV. There is a dominant edge effect noticed for the operating voltage beyond 30 kV with corresponding increase in SD at higher pressure and increased gap length. Unstable breakdown behavior beyond 30 kV is also substantiated with higher Δ that is observed

in the range of ~10 to 17 kV. The experimental data obtained for hemi-sphere electrode profile indicate that when SBV is less than 35 kV there is no noticeable variation in the SD and Δ also lies in the typical range of ~2 to 3 kV. Marginal increase in SD is observed as working voltage exceeds 35 kV for gap length of 2 mm and 3 mm. Due to radial expansion of *E* field on electrode surface area, Δ increased to ~10 to 12 kV. This effect on hemi-sphere profile was found to be gap dependent and it may be noticed that SD reduces with further increment in the gap length. This also supports the experimental observation corresponding to gap length of 4 mm and 5 mm as the data points were close to the mean value. Experimental results infer that hemi-sphere electrode profile will be the preferred choice for large discharge gap length (where higher hold-off voltage is needed) as the field uniformity inherently increases at higher gap lengths. The experimental results obtained for Bruce and Rogowski electrode profile are shown in Figure 14a,b.

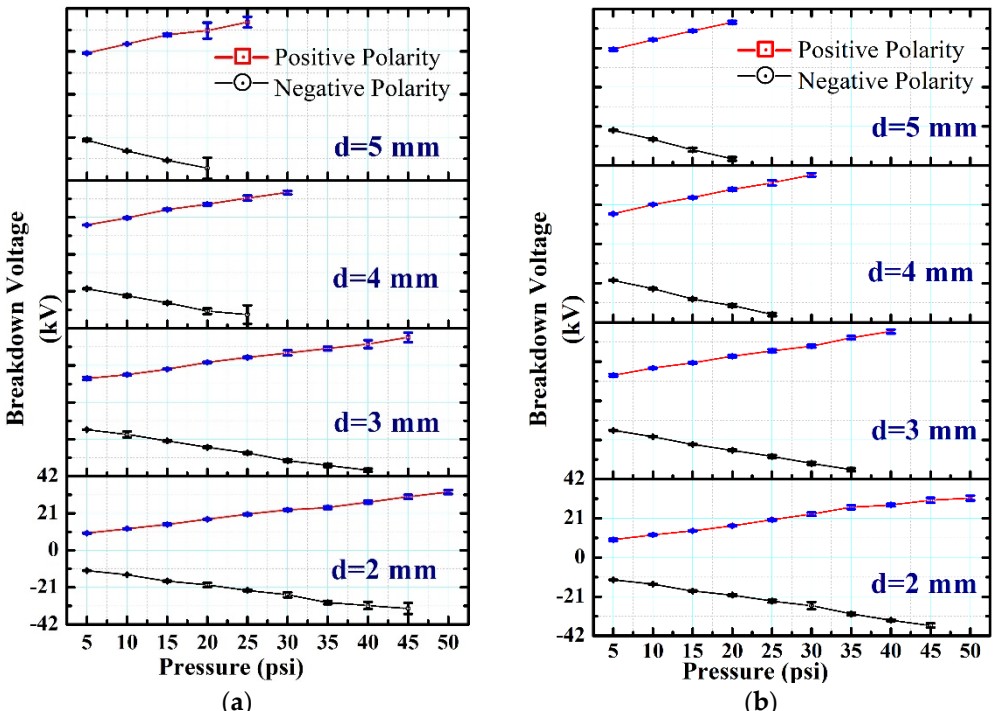

**Figure 14.** Results of Bruce and Rogowski profiles. (**a**) Bruce. (**b**) Rogowski.

Moderate increase in field uniformity factor η at subsequently higher gap lengths (as shown in Figure 8) has been evidently witnessed in experimentation with Bruce profile electrode. As shown in results in Figure 14a, at gap length of 2 mm, it efficiently holds the voltage up to ±35 kV while Δ was typically measured in the range of ~2 to 3 kV. As the distance between electrodes increased, field uniformity started deteriorating (unlike hemi-sphere profile). It may be noticed from the shown results that at higher gap lengths of 4 mm and 5 mm, hold-off voltage of Bruce profile electrodes reduced to 30 kV (mean value) while Δ typically increased in the range of ~8 to 16 kV. Subsequent increase in SD at higher gap length and increased pressures is an indication of gradual decrease in field uniformity.

Higher stability in steady state breakdown characteristics of Rogowski profile electrodes is evident from the experimental results shown in Figure 14b. The analytical observation reported in Section III and electrode profile characteristics shown in Figure 8 confirmed that there is very marginal change in field uniformity even at higher gap lengths. Rogowski profile electrode revealed stable high voltage hold-off up to ±40 kV over the gap lengths ranging between 2 and 5 mm with minimum Δ typically in the range of ~2 to 3 kV.

Another common observation in data of all profiles is that Δ has increasing trend at higher pressures for all inter-gap lengths. This is because for initiating breakdown at higher gas pressure, higher magnitude of voltage is applied and due to pronounced field

enhancement effects statistical variation in BDV is large. It is remarkable to note that field enhancement factor proportionately scales up with operating voltage and therefore in either condition, when inter-gap length is more or gas pressure is high, in both the cases Δ also increases.

Discharge signatures obtained on various electrode profiles after experimentation at gap lengths of 2 mm, 3 mm, 4 mm, and 5 mm are shown in Figure 15. Obtained discharge signatures on the electrodes found a good agreement with analysis and experimental findings reported in this paper. Signatures of erratic discharges extended over larger radii of Plane and Bruce profile electrode evidence detrimental effect on field uniformity at extended gap lengths. Confined discharge signatures over limited radii on electrode surface of Hemi-sphere and Rogowski profile electrodes infer much superior consistency in field uniformity even at higher gap lengths due to which these two electrode profiles show stable breakdown characteristics even at higher hold-off voltages and pressures.

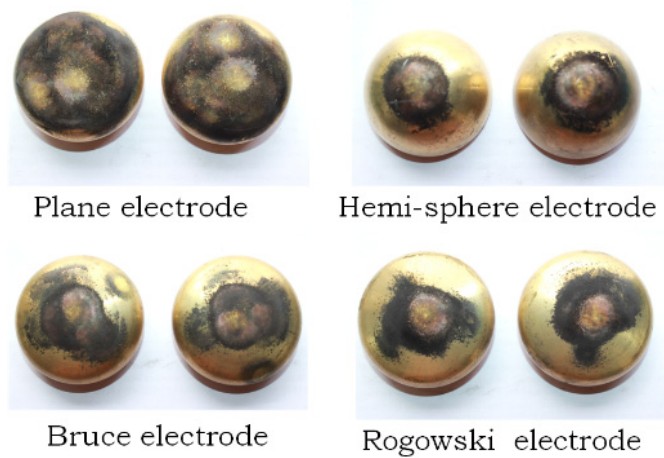

**Figure 15.** Discharge signatures on various electrode profiles.

In this experimentations, effect of polarity in hold-off voltage/SBV on respective electrode profiles has also been witnessed. From the experimental results shown in Figures 13 and 14 it is clearly evident that in all cases, SBV was found to be higher with negative polarity than positive polarity. However, the magnitude of difference (ranging between 2 to 6 kV) was found to be pressure dependent. The major cause behind this polarity effect is the presence of space charge in the discharge gap [16,39]. The space charge behavior is dominated by the characteristics of the filled gas. Contributing factors for the presence of space charge are geometrical errors, angular or axial misalignment in the electrode assembly, and localized field enhancement due to protrusions.

Analysis of time-resolved measurements in breakdown delay with positive and negative polarity revealed that switch closure is marginally fast with positive bias ($V_1$, $I_1$). Sample voltage and current discharge waveforms obtained for positive and negative polarity ($V_2$, $I_2$) at about ±33 kV are collectively shown in Figure 16.

The evident time delay of ~50 ns as a consequence of negative polarity is because the formative time-lag component has functional dependence on the breakdown mechanism which is explained by the streamer theory [40]. It is remarkable to note that the average propagation velocity differs for positive and the negative streamer. Although positive streamers propagate with higher velocity than negative streamers.

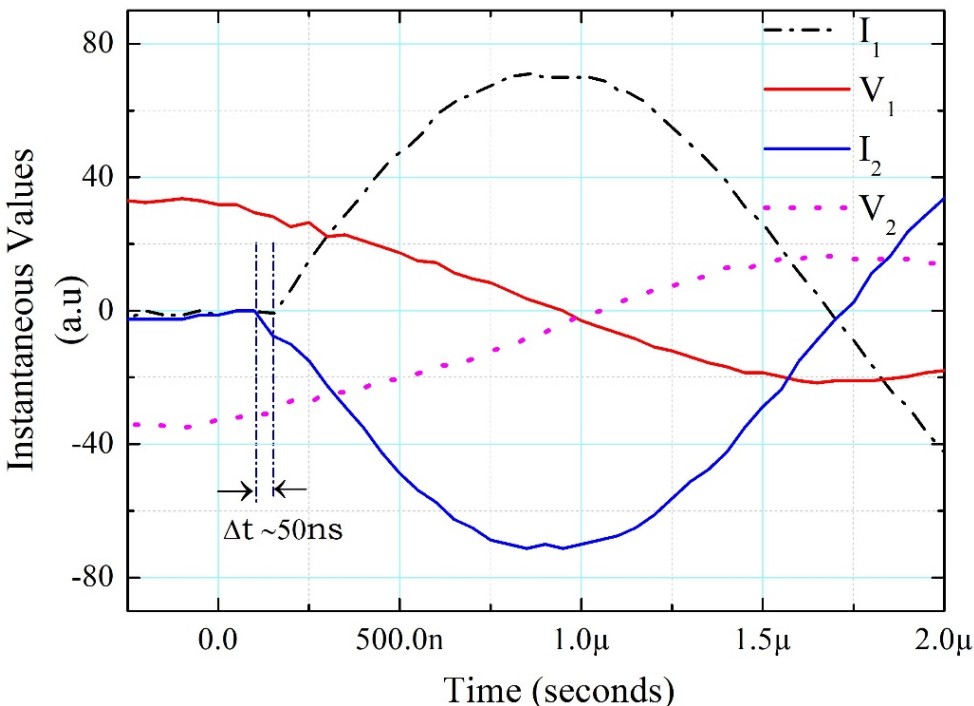

**Figure 16.** Voltage and current discharge waveforms with positive and negative polarity.

## 4. Conclusions

Owing to simplicity in design and ruggedness in use, sparkgap is the most commonly used pulsed power switch but obtaining reliable breakdown characteristics over wide working voltage range has always been a matter of important concern for practicing engineers. The investigation reported herein evidenced that stable hold-off voltage range and breakdown characteristic are majorly dominated by electrode dimensions and its profile. Thorough analysis, simulation, and experimentations performed on most commonly used four electrode profiles (Plane-parallel, Hemi-sphere, Bruce and Rogowski) revealed the following important findings:

- The analysis of spatial electric field distribution provides a generalized criterion for choosing major dimensions of sparkgap electrode and gap length to ensure uniform field in the inter-electrode region. Experimental investigation performed on four different types of electrode profiles mainly reveals the comparison of stability in breakdown characteristics of each profile so that their suitability is independently judged for use in appropriate operating voltage range.
- To ensure uniform electric field in the discharge gap region $d/t \leq 0.5$ (this results in a stable breakdown behavior as electric field is more focused in the discharge region rather than at edges).
- The variation of field uniformity factor $\eta$ with inter-electrode gap length $d$ dictates the breakdown behavior of sparkgap. With increase in the gap length, very steep raise, moderate change, and marginal variation in η is observed for plane parallel, Bruce, and Rogowski profiles respectively. As a result, even in the widest range of gap lengths Rogowski profile exhibits uniform electric field resulting in most stable breakdown behavior as compared to the remaining profiles.
- In Hemi-sphere electrode profile, $\eta$ improves with increase in gap length and hence it is a preferred choice in applications where large gap length is required to obtain very high hold-off voltage.
- Experimental investigation on the effect of polarity evidences higher SBV in all cases with negative polarity as a consequence of space charge build-up in the discharge gap. However, the extent of difference in SBV was found to be pressure dependent.

- In the time-resolved investigation of breakdown delay with positive and negative polarity it was recurrently noticed that switch closure is marginally fast with positive bias as a consequence of higher propagation velocity of positive streamer than the negative streamer.

The numerical simulations of avalanche evolution and its transition into cathode and anode directed streamers for aforementioned uniform field electrode profile will be the future scope of the investigation.

**Author Contributions:** Conceptualization, V.K.G. and R.V.; Investigation, writing-original draft preparation, V.K.G.; review and editing, R.V., M.W. and A.S. All authors have read and agreed to the published version of the manuscript.

**Funding:** This work was supported by the Bhabha Atomic Research Centre, Department of Atomic Energy, Government of India.

**Acknowledgments:** Authors are grateful to Shri Manraj Meena, Bijaya Lakshmi Sethi, and Shri Lakshman Rao Rongali for their kind help. One of the authors (G. Vinod Kumar) would like to thank Department of Atomic Energy (DAE), Government of India for financial assistance and support under DAE Graduate Fellowship Scheme 2018.

**Conflicts of Interest:** The author declare no conflict of interest.

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
