# Peer review of "Effect of Electrode Profile and Polarity on Performance of Pressurized Sparkgap Switch"

_plasma, doi:10.3390/plasma5010010_

Round 1

Reviewer 1 Report

1) The literature review is relatively short and insufficient (there should be no less than 30 sources).
2) The novelty of the work should complete the introduction.
3) the proposed novelty should have the countable practical significance mentioned at the conclusions.
4) The structure of the article should be classical.
5) The conclusions should be short and countable. The first conclusion should be on achieving the main aim of the study. Please add further outlook as well.
6) The current version should be improved before publication.

Author Response

Respected Sir,

                       The Manuscript has been revised as suggested. In comment 5 stated conclusions should be  short and countable. The conclusions section has been re-written but there are number of conclusions and each of them has its own significance. please, consider my request. 

Reviewer 2 Report

I like the paper because it includes, Theory, Simulation and Experimental data from a prototype. The abstract introduces the reader to important findings. There is a nice discussion comparing simulated and experimental data.

However, there is room for improvements.

  1. The paper is not written by 1 person. Only section 5 of the paper is well written with proper use of the third person. Please, use 3rd person throughout the paper.
  2. Typing mistakes. Please proof read and correct.
    • Line 252: The Evident..... e should be lower case.
    • Line 110: The results infer not infers.
    • Line 257: Conclusions not Conlusion's
    • There are more.
  3. Equations: Font size and styles are different.
  4. Figure axis-labels use different font size and styles.
  5. Figures 13 and 14 are confusing because the red line has blue dots and the black line has red dots whereas the legend is different.

Thank you.
